# Inducing Osteogenesis in Human Pulp Stem Cells Cultured on Nano-Hydroxyapatite and Naringin-Coated 3D-Printed Poly Lactic Acid Scaffolds

**DOI:** 10.3390/polym17050596

**Published:** 2025-02-24

**Authors:** Reem Mones Dawood, Anas Falah Mahdee

**Affiliations:** Restorative and Aesthetic Dentistry Department, College of Dentistry, University of Baghdad, Baghdad 1417, Iraq; a.f.mahde@codental.uobaghdad.edu.iq

**Keywords:** regeneration, pulp dentin complex, vital pulp therapy, tooth revascularization, stem cells

## Abstract

Background: Regeneration dentistry demonstrates significant challenges due to the complexity of different dental structures. This study aimed to investigate osteogenic differentiation of human pulp stem cells (hDPSCs) cultured on a 3D-printed poly lactic acid (PLA) scaffold coated with nano-hydroxyapatite (nHA) and naringin (NAR) as a model for a dental regenerative. Methods: PLA scaffolds were 3D printed into circular discs (10 × 1 mm) and coated with nHA, NAR, or both. Scaffolds were cultured with hDPTCs to identify cellular morphological changes and adhesion over incubation periods of 3, 7, and 21 days using SEM. Then, the osteogenic potential of PLA, PLA/nHA/NAR, or PLA scaffolds coated with MTA elutes (PLA/MTA scaffolds) were evaluated by measuring mineralized tissue deposition using calcium concentration assays and alizarin red staining (ARS). Also, immunofluorescence labelling of alkaline phosphatase (ALP) and dentine sialophosphoprotein (DSPP) within cultured cells were evaluated. Results: The highest cellular attachment was identified on the PLA/nHA/NAR scaffold, with morphological changes reflecting cellular differentiation. The highest calcium deposition and ARS were recognized in the PLA/nHA/NAR culture, with statistically significant difference (*p* < 0.05) compared to PLA/MTA. Also, ALP and DSPP markers showed statistically significantly higher (*p* < 0.05) immunoreactivity in cells cultured within PLA/nHA/NAR compared to PLA/MTA. Conclusions: The results confirm the osteogenic potential of PLA scaffolds coated with nHA/NAR for future animal and human investigations.

## 1. Introduction

Tissue engineering is a critical approach aimed at restoring damaged or missing tissues by harnessing the body’s inherent healing mechanisms. Key elements of this process are various cell types, in particular mesenchyme stem cells (MSCs), that are characterized by their ability to self-renew [1] and their multipotency [2], which allows them to differentiate into osteoblasts, smooth muscle cells, chondrocytes, and adipocytes [2]. The major challenge in this approach is how to control the proliferation and differentiation into the desired somatic cells and how to sustain the resulting differentiated phenotype [3]. In pulp regeneration, these obstacles can be vanquished through adequate scaffold type and design to control stem cell differentiation into a pulp dentin complex [4]. Three-dimensional printing technology is a controllable manufacturing process that utilizes a diverse array of biocompatible materials offering innovative solutions toward this aim [5].

Different synthetic and natural materials have been used for fabrication of 3D-printed scaffolds; for example, poly(lactic-co-glycolic acid), poly lactic acid (PLA), polycaprolactone, gelatin, collagen, chitosan, hyaluronic acid, and alginate are the most frequently employed. Although natural scaffolds can provide better cell adhesion, with more hydrophilic structure, synthetic scaffolds can be synthesized with a controlled rate of degradation and higher mechanical properties.

PLA is a biodegradable polymer that has acceptable mechanical properties and modulus of elasticity and which has been used as a scaffold in tissue engineering. However, this material has poor surface adherence due to the lack of surface cell discrimination points, with no hydrophilicity and cellular affinity [6]. Therefore, it requires coating with a hydrophilic hydrogel such as poly-vinyl alcohol (PVA) to make its surface prone to receive bioactive material. This may facilitate attachment of stem cells on the scaffold surface to promote cellular proliferation and differentiation [7,8,9].

Several bioactive compounds such as naringin (NAR) and nano-hydroxyapatite (nHA) are known to elicit specific biological responses that promote cultured cell adhesion, proliferation, and differentiation. Naringin, a flavonoid compound extracted from grapefruit, tomatoes, and citrus fruits [10], has potent bioactivity promoting human mesenchyme stem cell proliferation and differentiation [11,12]. Nano-hydroxyapatite (nHA), the main constituent of hard tissue, has confirmed biocompatibility, structural stability, and excellent bone conductivity as demonstrated through several tissue engineering investigations [13,14,15]. Furthermore, studies have found that the combination of nHA/NAR exhibits boosted bioactivity, osteoconductivity, and biocompatibility, making it an ideal candidate for dental tissue engineering. They can effectively support the growth and differentiation of stem cells, such as dental pulp stem cells (DPSCs), enhancing their potential for clinical applications in bone regeneration [16].

A new scaffold model for pulp regeneration was suggested in our previous work [17]. This model is based on using low-cost and easily manufactured 3D-printed PLA scaffolds with two pore sizes (300 and 700 µm), and surface coating them with bioactive compounds (nHA, NAR, and a combination of the two). Results showed that a PLA scaffold with a pore size of 300 µm exhibited superior physical properties, and a combined coating of nHA and NAR could produce an active surface layer which effectively increased the antibacterial and bioactivity of PLA scaffold. Therefore, the current study aims to evaluate cellular attachment and differentiation in the culturing of the suggested scaffold model compared to MTA-treated scaffolds, as MTA has established itself as a reliable material in endodontics with significant advantages for pulp regeneration, promoting cellular attachment due to its biocompatibility and favorable surface interaction properties [18,19]. The cellular osteo/odontogenic differentiation is investigated by measuring calcified nodule formation through calcium ion deposition and alizarin red staining. In addition, the alkaline phosphatase enzyme (ALP) and dentine sialophosphoprotein (DSPP) levels were identified within the cultured cells using an immunolabeling technique. These proposed scaffolds can be used for vital pulp therapy and revascularization treatments, but warrant further investigation.

## 2. Materials and Methods

### 2.1. Fabrication of the 3D Printed PLA Scaffold

Scaffold manufacturing and dipping procedures are based on our previous work [17]. For the fabrication of 3D-printed PLA scaffolds, an FDM-based 3D printer from Creality 3D Technology Co., Shenzhen, China, was utilized. This printer operated using a 1.75 mm diameter PLA filament and featured a nozzle diameter of 200 µm. The objective was to create scaffolds with a multilayered porous microstructure characterized by tubular pore shapes surrounded by a denser perimeter.

The resulting scaffolds were designed as multilayered, 300 µm porous structures in the form of circular sheets, measuring 10 mm in diameter and 1 mm in height (as shown in Figure 1). The specific parameters employed during the fused deposition modeling (FDM) printing technique are detailed in the study (Table 1). These parameters were essential for achieving the desired porous microstructure and dimensions of the PLA scaffolds.

A simple dip coating method was employed to prepare four distinct groups of 3D-printed PLA scaffolds (neat PLA, PLA/nHA, PLA/NAR, PLA/nHA/NAR). For effective bonding of the bioactive components to the scaffold surface, 1 g of analytical grade PVA was added to 50 mL of distilled water and stirred using a hot plate magnetic stirrer at 50 °C until a uniform solution was formed [20]. Subsequently, 0.1 g of (nHA) (HAP01, Hualan Chemistry^®^, Xinxiang, China) was incorporated into the PVA solution, and the resultant mixture was adjusted to a total volume of 100 mL before being subjected to ultra-sonication at 40% amplitude for 120 s to ensure thorough dispersion and refinement of the nano-hydroxyapatite particles [21]. The same method was applied for the NAR solution (Sigma Aldrich, Co., St. Louis, MO, USA). For the combined nHA/NAR formulation, the NAR solution was gradually added to the nHA solution using a dropper while continuously stirring for an additional 30 min before undergoing sonication for 120 s.

Coating was achieved by soaking PLA samples in the chosen mixture for 1 min and drying with hot air for 5 min per dipping cycle, which was repeated for five times. All coated PLA samples were left to dry in a desiccator at room temperature for 24 h [22].

### 2.2. In Vitro Cell Culture

#### Preparation and Proliferation of Human Dental Pulp Stem Cells (hDPSC)

This study obtained ethical approval from the ethics committee in the College of Dentistry, University of Baghdad, with reference number 760 on 12 January 2023.

Dental pulp stem cells (DPSCs, ATCC: C555) were procured from the National Center of Genetic Resources in Iran. The cells were cultured in a 96-well plate containing Dulbecco’s Modified Eagle Medium (DMEM) (GIBCO, Grand Island, NY, USA), supplemented with 10% fetal bovine serum (FBS), 1% L-glutamine, 100 U/mL penicillin, and 100 μg/mL streptomycin (GIBCO, USA). The cells were incubated at 37 °C in a humidified atmosphere with 5% CO_2_, with medium changes every 2 to 3 days to maintain optimal growth conditions, following the ATCC culturing protocol for DPCs. After sub culturing, the cells were detached, and harvested using Trypsin-EDTA (Sigma-Aldrich, Taufkirchen, Germany) solution at a concentration of 0.25% trypsin with 0.53 mM EDTA, as specified in Sigma’s standard protocol for adherent cell lines [23], for 5 min at 37 °C once they reached 70–80% confluence. In this study, the cells were expanded to passage 4 to be used for all experiments. This was to obtain healthy cell characteristics and minimize variability due to genetic drift or altered behavior associated with higher passage numbers [24].

### 2.3. Cell Adhesion to the Scaffold

To evaluate the biocompatibility of the fabricated scaffolds, a cell proliferation test was conducted, with analyses performed in triplicate at the end of each specified time point. For this evaluation, twelve PLA scaffolds were created to fit into 24-well plates, each with a height of 1 mm and a diameter of 16 mm. This design was intended to prevent the floating of the scaffolds during cell culture experiments. The fabricated scaffolds were sterilized by UV, then categorized into four groups based on the treatments received (PLA/nHA, PLA/NAR, PLA/nHA/NAR, and neat PLA as control without treatment). Scaffolds from each group were soaked in 300 µL of fetal bovine serum (FBS) (GIBCO-USA) for 24 h to enhance cell attachment [25]. Following this, the scaffolds were placed in cell culture plates to be conditioned in a nature-inspired culture environment for 3 h at 37 °C [26].

The cultured scaffolds (neat PLA, PLA/nHA, PLA/NAR, PLA/nHA/NAR) were then coated with 100 μL of complete culture medium containing an aliquot of 1 × 10^4^ hDPSCs. These were incubated for 2 h at 37 °C in a 5% CO_2_ atmosphere to facilitate initial cell attachment [24,26]. Then, scaffolds were fully covered with the previously prepared complete culture medium [24], and incubated for three time intervals (3, 7, and 14 days).

Cell attachment to the scaffolds was evaluated using scanning electron microscopy (SEM). After each incubation period, the scaffolds were removed and gently rinsed twice with phosphate-buffered saline PBS, to remove non-adherent cells or debris. Scaffolds werethen fixed with 2.5% (*w*/*v*) glutaraldehyde (Merck, KGaA, Darmstadt, Germany) at 4 °C for two hours [27]. Following three phosphate-buffered saline (PBS) rinses, the specimens were subsequently dehydrated using an ascending series of alcohol solutions (25%, 50%, 75%, 96%, and 100% ethanol), with 15 min in each [28]. The specimens were then allowed to air dry; subsequently, a thin layer of approximately 10 nm of gold was applied via sputtering for 30 s [29] to prepare them for analysis via scanning electron microscopy (SEM) (Hitachi, Tokyo, Japan) to observe and assess cell morphology and adhesion [28,30,31].

### 2.4. Osteogenic Markers

The osteo-inductive properties of the scaffolds were demonstrated by matrix mineralization observed within hDPSC culture on the scaffold at day 21. For this purpose, the cells were maintained in a growth medium without osteogenic supplements, such as dexamethasone, ascorbic acid, or β-glycerophosphate [32], to confirm that the observed osteogenic differentiation was attributed solely to the interactions between cells and the treated scaffold biomaterial, without additional osteoinduction factors. To assess early osteogenic differentiation, calcium mineral deposition and the scaffold’s overall preferences for osteogenic or odontogenic differentiation of the attached cells were evaluated [32]. 

### 2.5. Calcium Concentration Assay

Based on the results of the best cell adhesion scaffolds, only four different groups were used for measuring calcium deposits, analyzed in triplicates (n = 3): control group (cultured in well plate), PLA (without coating), PLA/nHA/NAR scaffolds, and PLA/MTA scaffolds. The latter scaffold was prepared by mixing 0.1 g of MTA white bio-ceramic repair material (PD, Vevey, Switzerland) with 50 mL of PVA solution, which was then adjusted to a total volume of 100 mL with distilled water. This mixture underwent ultra-sonication at 40% amplitude for 120 s and plain PLA scaffolds were dip-coated with this solution as previously described.

To detach cells from the culture well (control group), initially, completely aspirate the existing cell culture medium. Next, add 300 µL of phosphate-buffered saline (PBS) to the well for washing, to help in removing any residual serum components. Then, use a sterile cell scraper to gently dislodge the cells from the bottom of the well. After scraping, the cell suspension was aspirated using the reverse pipetting technique to ensure that all detached cells were aspirated without causing additional stress [33], Finally, the collected cell suspension was centrifuged at a low speed of 300× *g* for 5 min to pellet the cells for further analysis [33]. In a separate process for scaffolds, they were taken from their designated culture plates after 21 days of incubation in cell culture medium, and washed with cold phosphate-buffered saline (PBS) (0.01 M, pH 7.4). They were then homogenized in 180 µL of deionized water using a Dounce homogenizer at 4 °C, then centrifuged at 10,000× *g* for 10 min to remove insoluble material, and then the supernatants were collected [34]. The calcium content in these supernatants was determined using a colorimetric assay (Elabscience Bionovation Inc., Houston, TX, USA) according to Elabscience Calcium Concentration Assay protocol with a micro plate reader (Bio Tek, Instruments, Winooski, VT, USA), set to a wavelength of 610 nm. The absorbance values were converted to calcium concentrations in mMol/µL using a standard calcium curve, following the manufacturer’s instructions (Figure 2B).

### 2.6. Alizarin Red Staining and Oseoconductivity of the Scaffold

Following a similar culturing procedure as in previous tests, neat PLA scaffolds, PLA/nHA/NAR, and PLA/MTA were cultured in 24-well plates with triplicates. After a 21-day culture period, the culturing media were removed, and scaffolds were fixed with 1 mL of 4% paraformaldehyde (Sigma) added to each well for 20 min, followed by three washes with PBS for 5 min each. Subsequently, 1 mL of 2% alizarin red staining solution was added to each well. This solution was prepared by dissolving 2 g of alizarin red (Merck, Rahway, NJ, USA) in 100 milliliters of distilled water, with the pH adjusted to 4.1–4.3 using 10% ammonium hydroxide. The staining incubation lasted between 1 to 5 min at room temperature [35], with a characteristic red-orange color indicating calcium deposits typically appearing after just 2 min. To eliminate excess dye and prevent non-specific staining, samples were washed with 100% ethanol and then rinsed three times with PBS. Finally, imaging was conducted using a light microscope (LABOMED, Los Angeles, CA, USA).

For the quantification of Alizarin Red S, the stain was extracted from the cultured scaffolds using 200 µL of 10% acetic acid solution per well [36], which was incubated for 30 min under gentle shaking [37]. The resulting solution was transferred to an Eppendorf tube and heated at 75 °C for 10 min. The mixture was then centrifuged at 12,000 rpm for 15 min to separate the components. From this process, 100 µL of the supernatant was collected for absorbance measurements at a wavelength of 450 nm [38] using a microplate reader (Bio Tek, USA).

### 2.7. Alkaline Phosphatase (ALP), and Dentine Sialophosphoprotein (DSPP) Expression

Immunofluorescence staining was carried out at day 21 of culture to detect ALP and DSPP. The expression of these proteins was investigated in hDPSCs grown on neat PLA scaffold, PLA/nHA/NAR, and PLA/MTA. Experiments were carried out in triplicate. After cellular fixation as mentioned previously, scaffolds were washed 3 times in PBS, then treated for 10 min with 0.1% Triton X-100 at room temperature, and washed again 3 times with PBS [39].

Scaffolds were then incubated with blocking solution (1% BSA (bovine serum albumin) in PBS) at room temperature for 1 h to block unspecific binding of antibodies during immunofluorescence staining [40]. The blocking solution was removed before adding the primary antibody. Mouse monoclonal anti-TNAP (Biorbyt, Ltd., Cambridge, UK), rabbit polyclonal anti-DSPP (Biorbyt, Ltd., Cambridge, UK) antibodies were used for the detection of ALP and DSPP, respectively, at 1:100 dilution in PBS, incubated for 1 h at room temperature [39]. After 3 PBS washes, scaffolds were incubated for 1 h at room temperature with Goat anti-rabbit polyclonal IgG(H + L) (Biorbyt, Ltd., Cambridge, UK), highly cross-adsorbed secondary antibody conjugated with fluorescein isothiocyanate, (FITC) (Biorbyt, Ltd., Cambridge, UK) [39]. Nuclei of differentiated cells were stained with 0.5 mg/mL of 4,6-diamino-2-phenylindole (DAPI) [41,42].

Automated quantification of the fluorescence intensity was adapted, and large-field fluorescence images were captured using a Leica LAS X Widefield System (Leica Microsystems GmbH, Ernst-Leitz-Strasse 17-37, Germany) and ImageJ software (version 1.42 g) [43,44] and analyzed under fixed settings of brightness, contrast, and an image magnification of 40× Relative fluorescence intensity was calculated by measuring the sum of each pixel value (integrated density) above a threshold of 10,000 divided by the overall number of cells obtained in phase-contrast images [45]. The rolling ball algorithm provided by the ImageJ software was employed to subtract backgrounds and for correction of uneven illumination, preserving low-intensity details in our captured images [46,47].

### 2.8. Statistical Analysis

GraphPad Prism 6 (GraphPad Software, San Diego, CA, USA) was used to conduct the statistical analysis, after identifying data normality by using the Shapiro–Wilk test. Quantitative measures for all assays, including calcium concentration assay, alizarin red staining assay, alkaline phosphatase (ALP), and dentine sialophosphoprotein (DSPP) immunofluorescence assays were presented as mean ± standard deviation. One-way ANOVA was employed to evaluate statistical differences among (2D cell culture, neat PLA, PLA/nHA/NAR, and PLA/MTA) groups at a significance level of *p* < 0.05. Then Tukey’s multiple comparisons test was used to identify specific differences amongst the groups. All statistical tests were performed in accordance with standard practices for assessing variances across groups in biological assays.

## 3. Results

### 3.1. Cell Adhesion

The suitability of the 3D structure of the scaffolds for cell seeding was assessed by observing cell morphology and adhesion using SEM imaging, as it plays a pivotal role in promoting osseointegration. Figure 3 presents SEM microphotographs of scaffolds taken as early as the third day, demonstrating cellular adherence on all treated scaffolds, with less common attachment results observed on both neat PLA and PLA/NAR scaffolds. SEM images show that the attached cells exhibited a spindle-shaped fibroblast-like appearance with long cytoplasmic processes, which was most prominent in the PLA/nHA/NAR group, suggesting that this combined surface was effective in promoting cell adhesion, thus exhibiting better biocompatibility. Therefore, this scaffold was chosen for further osteo-odontogenic evaluations.

**Calcium concentration assay:** Figure 2A illustrates that the amount of calcium deposition in the PLA/nHA/NAR group was significantly higher than that in the PLA/MTA group (*p* = 0.012), indicating a stronger potential for bone mineralization and cell differentiation in this group. This result suggests that the bioactivity of the PLA/nHA/NAR combination in promoting osteo-odontogenic differentiation is more prominent.

**Alizarin red staining:** Images in Figure 4A demonstrate that scaffolds coated with biomaterial (PLA/nHA/NAR, and PLA/MTA) exhibited the highest intensity of alizarin red staining. This indicates that these scaffolds induced mineral matrix deposition more than the control and uncoated scaffold. Quantification was performed by eluting alizarin red staining and acquiring optical density measurements (Figure 4B). Results showed that the mineralization rate in the control and PLA groups, which had mean values of 30.7 ± 4.61 and 36.3 ± 3.53, respectively, were statistically significantly lower than those observed in theother groups. Notably, the PLA/nHA/NAR scaffold demonstrated a significantly higher optical density of alizarin red staining than the PLA/MTA scaffold (*p* = 0.001) (see Figure 4B).

### 3.2. Immunofluorescence Staining

The results of immunofluorescence staining showed that ALP and DSPP were significantly immunoreactive in the cytoplasm of the PLA/nHA/NAR and PLA/MTA groups, indicating strong expression of these two markers, as they are related to the osteogenic/odontogenic processes of the cells. Compared to the control and neat PLA groups, where they illustrated very low cellular expression of both markers, the expression levels of ALP and DSPP in the PLA/nHA/NAR group were significantly higher than PLA/MTA (*p* < 0.03), indicating that it has a role in promoting mineralization and dentin/bone differentiation and also indicating stronger biological activity (Figure 5).

## 4. Discussion

The regeneration of the dental pulp complex is a critical area of focus in endodontics, particularly amid the challenges associated with dental pulp injuries. These injuries result in significant pulp damage, compromising tooth vitality, and lead to critical clinical challenges and loss of tooth vitality, prompting the need for innovative regenerative strategies. The suggested model of a 3D-printed poly lactic acid (PLA) scaffold coated with nHA and naringin can support cellular attachment of hDPSC, and induce their osteo/odonto- differentiation into calcified tissue-forming cells. This is supported by positive outcomes from alkaline phosphatase (ALP) and dentin sialophosphoprotein (DSPP) assays. Therefore, this model can offer a promising avenue for enhancing pulp regeneration and improving clinical outcomes.

In the present study, a 3D printing technique was employed for scaffold fabrication as it is capable of forming customized macro structures with delicate intricate interconnections. Polymeric scaffolds with micro porosities of (300 μm) were employed as suggested by the literature [48]. These micro-pores can enhance cell seeding, and increase cell density and cell-to-cell communication [49]. Another study suggested that the range of pore diameter could reach up to 1000 µm and still be suitable for bone tissue engineering. As it could provide more open structures and growth areas to capture and induce cell ingrowth [50]. This controversy could be due to differences in the scaffold design, the material used in its fabrication, methods of fabrication, and the origin of the stem cells used in different studies [51,52].

While poly lactic acid (PLA) scaffolds have been widely used in tissue engineering due to their biocompatibility and biodegradability, they exhibit inherent hydrophobic properties. This can hinder cellular adhesion, which is crucial for successful tissue regeneration [53]. To address this limitation, the enhancement of PLA scaffold surface with bioactive coatings is mandatory. These coatings can improve its hydrophilicity for better cell attachment and promote the cellular activities necessary for tissue formation [54]. Therefore, polyvinyl alcohol (PVA) was utilized as a binding agent [7,54] along with NAR and nHA as bioactive coating materials [17].

Cell attachment is an essential prerequisite in tissue regeneration, as it directly influences cell viability, proliferation, and differentiation [55]. Initial adhesion of cells to a scaffold initiates a cascade of biological events that determine cell fate, including the formation of local adhesions and the activation of signaling pathways that promote subsequent cellular activities In this study, the cultured hDPSCs were apparently attached to the scaffold surface and exhibited an initial spherical, and more rounded appearance. This finding aligns with results from previous studies [24,28,56,57,58].

The highest levels of cell attachment were observed in PLA scaffolds that incorporated both nHA and NAR. This may reflect the synergistic effect of this combination, which results in a scaffold with an optimal microenvironment for dental pulp stem cells (hDPSCs), leading to significantly higher rates of cell attachment and subsequent cellular activities compared to other tested scaffolds. This agrees with a study by Zhang et al. in 2022, which illustrated that combining nHA and collagen followed by the encapsulation of NAR can enhance cellular attachment, proliferation, and osteogenic differentiation [16].

In the current study, two scaffold coatings were investigated for their effect on calcium ion release and overall bioactivity. The results indicated that both PLA/nHA/NAR and PLA/MTA significantly enhance the odontogenic differentiation of the hDPSCs at a higher rate compared to the control and neat PLA scaffold. Furthermore, it was evident that PLA/nHA/NAR exhibited a greater potential for oseo-odontogenic differentiation compared to PLA/MTA. These results could be attributed to several factors; notably, the chemical composition of the bioactive coating material. The hydroxyapatite is a naturally occurring calcium phosphate, which is an analogous chemical composition to the components of calcified tissues, including bone and teeth, mimicking their specific crystal arrangement [59,60]. Moreover, the nanoscale of hydroxyapatite particles may significantly enhance the osteogenic protein expression, thereby promoting mineralization and cellular differentiation [14].

On the other hand, naringin (NAR), a natural flavonoid compound, has been shown to activate the Wnt/β-catenin signaling pathway, which plays an important regulatory role in bone and tooth differentiation. By activating this pathway, NAR can upregulate the expression of osteogenic markers including DSPP, thereby enhancing the differentiation potential of hDPSCs. The literature shows that NAR regulates local pH through antioxidant effects, helps to increase the solubility of nHA, and then promotes the sustained release of calcium ions. This accelerates the bone formation process [61,62,63,64,65,66]. Additionally, studies have shown that incorporating NAR into our construct effectively leverages its potential to stimulate bone-forming cells (osteoblasts) and inhibit bone-resorbing cells (osteoclasts), thereby significantly enhancing overall bone regeneration [16].

The dual action of NAR in conjunction with nHA enhances the overall regenerative capacity of the presented formulation. This combination can create a synergistic environment that supports and accelerates mineralization processes. According to the literature, the antioxidant properties of NAR could modulate the local pH, thereby enhancing hydroxyapatite’s solubility. This can provide a sustained calcium ion release, which actively stimulates osteoblast differentiation over time [61,62,63,64,65,66].

In contrast, MTA-coated PLA scaffolds showed limited mineralization potential compared to the combination of nHA/NAR. This aligns with a study by Hanafy et al. (2018), which found that nHA can facilitate stem cell proliferation and differentiation more effectively than MTA, making it a promising candidate for regenerative dental applications [67]. This could be attributed to the hydration process of MTA, which can initially produce calcium silicate hydrate (C-S-H) and calcium hydroxide (Ca(OH)_2_). This results in a gradual release of calcium ions that decreases over time as the C-S-H disintegrates and the material matures [68]. Therefore, MTA does release a significant amount of calcium primarily, but it diminishes as hydration progresses [69], and this could lead to a lower overall availability of calcium ions compared to hydroxyapatite. However, to confirm this suggestion, we suggest a longer evaluation time to investigate the role of time between these coating materials.

In this study, the key innovation of the proposed scaffold lies in its three-dimensional (3D) structure, which provides a microenvironment for cellular infiltration, differentiation, and guided tissue regeneration. Its porous architecture helps stem cells to attach and proliferate within its structure, enhancing tissue integration. Also, its 3D framework could offer deeper cellular interactions, potentially reaching deeper into the induced tissue for a more biologically driven repair process. This is completely absent in the biological model offered by all the available pulp regenerative materials, which often depend on the capping mechanism rather than interlapping within the capped tissue [70]. In addition, the controlled and customized release of bioactive molecules can promote odontoblast differentiation and matrix mineralization. Thus, this 3D-printed model with bioactive coating materials needs further investigation to validate its applicability compared to conventional capping cement materials.

However, some limitations should be noted, including potential challenges in scaling up the 3D printing process, and ensuring uniform coating on complex structures. In addition, there is a critical need for further long-term studies on stability, degradation, and potential cytotoxicity of the coatings in vivo, through animal models, especially in terms of blood flow, immune response, and tissue integration. Additionally, while the cytotoxicity of NAR, nHA, and combinations of them were evaluated in our previous study [17], a long-term cytotoxicity assessment involving a diverse range of cell types may be necessary to comprehensively evaluate the broader safety profile and biocompatibility of these tested materials.

Overall, the PLA/nHA/NAR scaffold showed promising results for regenerative dentistry applications, but additional research is needed to address these limitations and compare its long-term performance to existing treatments.

## 5. Conclusions

This study highlights the significant potential of 3D-printed poly lactic acid scaffolds coated with nHA and NAR in enhancing osteo-odontogenic differentiation of cultured hDPSCs. The incorporation of these bioactive formulations showed improvement in cellular adhesion, enhancement in ALP enzyme formation, and increased mineralized tissue deposition. By harnessing these bioactive properties, the suggested innovative scaffold design presents a promising avenue for advancing dental tissue regeneration.

## Figures and Tables

**Figure 1 polymers-17-00596-f001:**
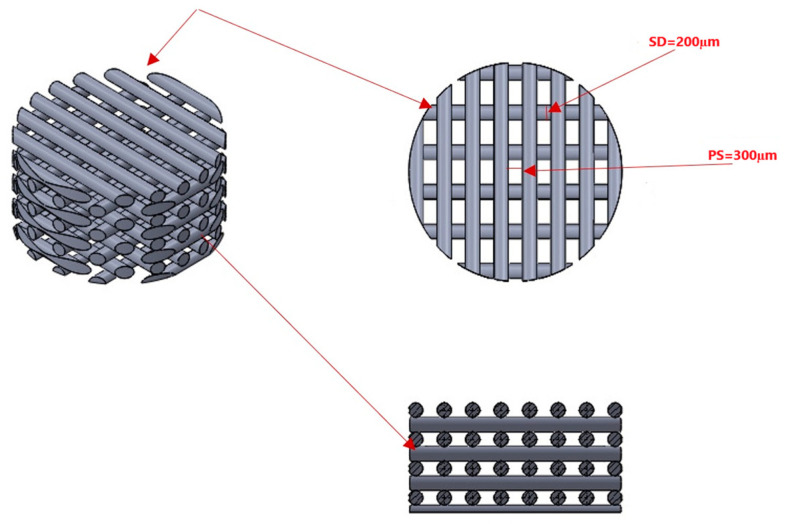
Design of 3D-printed scaffolds appears as circular and multilayer sheets (10 mm diameter × 1 mm height).

**Figure 2 polymers-17-00596-f002:**
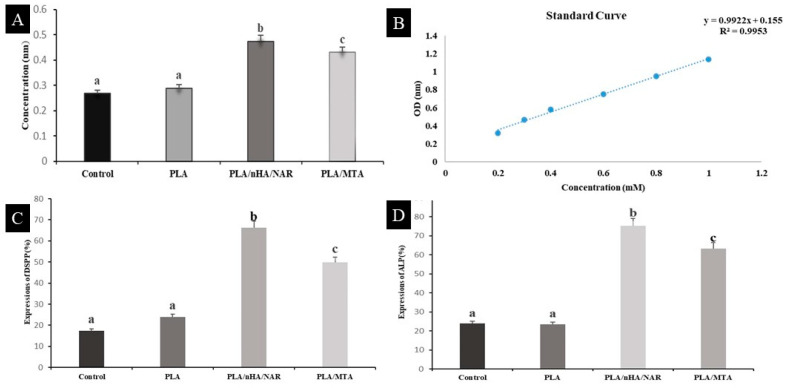
Bar charts for groups of different tests: (**A**) calcium concentration assay, (**B**) calibration curve of calcium ion, (**C**) expression percentages of immunofluorescence staining for DSPP, and (**D**) expression percentages of immunofluorescence staining for ALP. Different letters above bars represent statistical significance at *p* ≤ 0.05.

**Figure 3 polymers-17-00596-f003:**
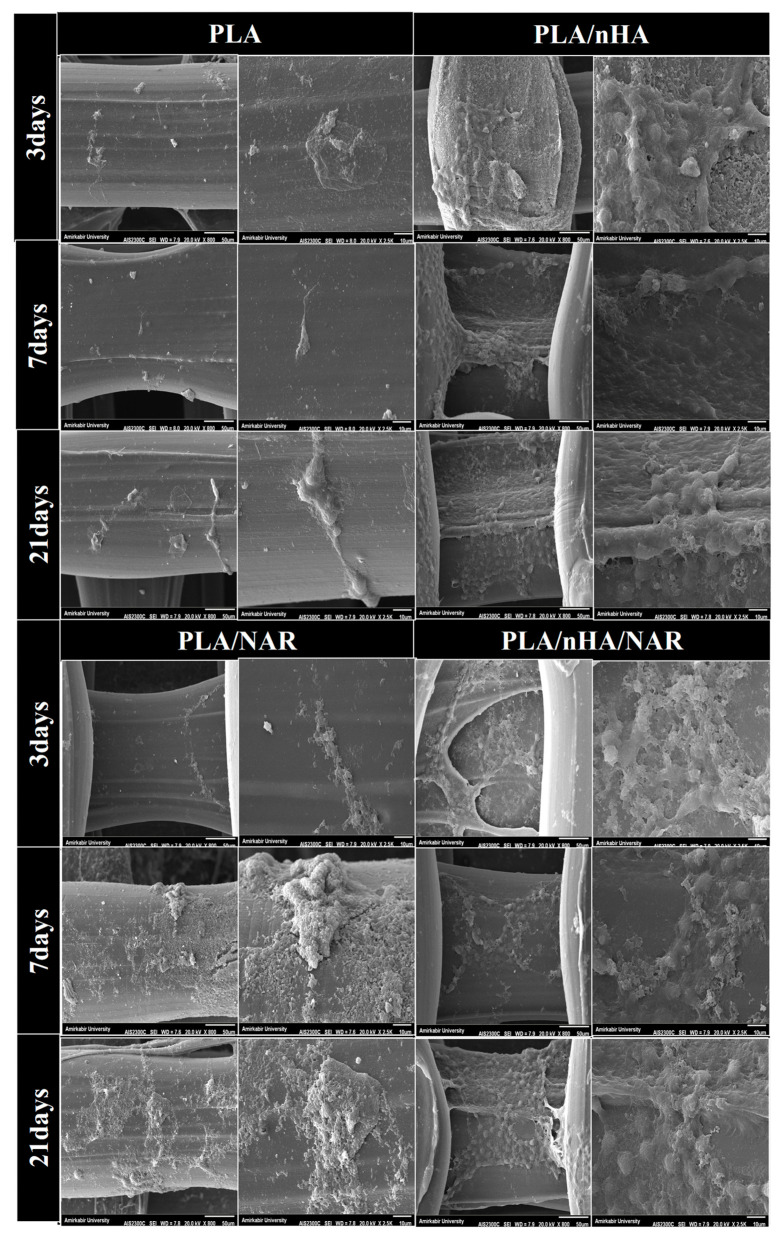
Representative micrographs of hDPSCs adhered on micro-patterned surfaces of PLA, PLA/nHA, PLA/NAR, and PLA/nHA/NAR, taken at 3, 7, and 21 incubation days. Micrographs of each group are arranged into two panels based on magnifications (the left panel shows images at a magnification of 800×, while the right panel shows images at a magnification of 2.5×).

**Figure 4 polymers-17-00596-f004:**
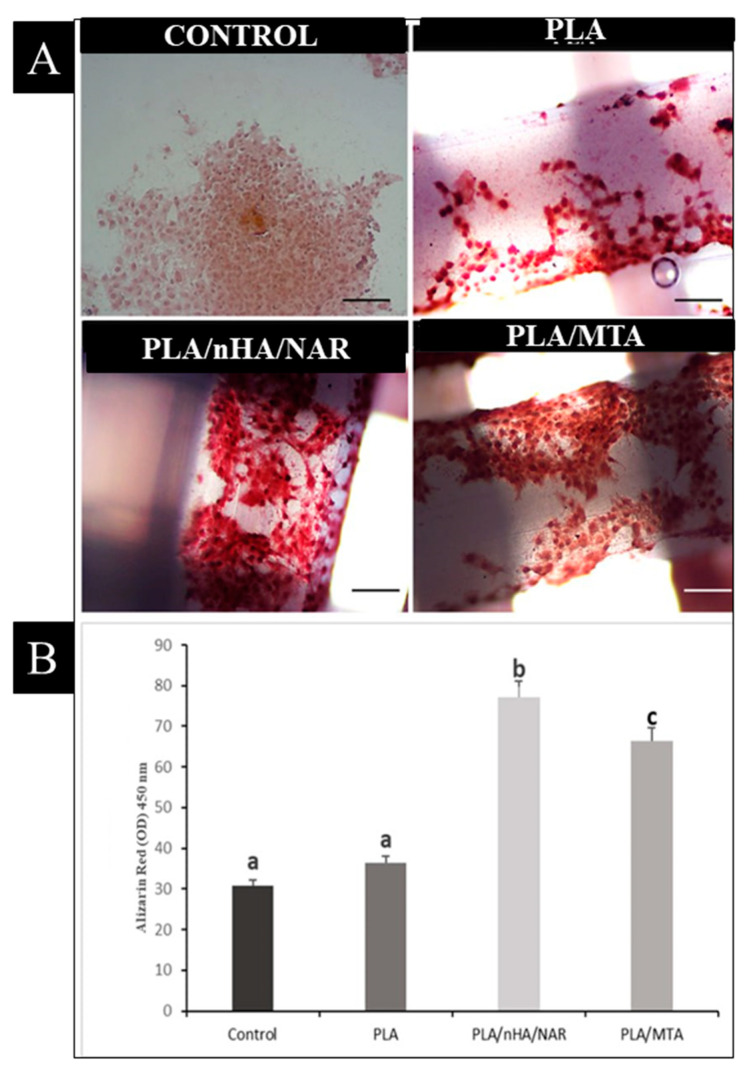
Alizarin red staining and optical density comparison for all groups. (**A**) Representative images of transmitted light microscopy at 20× magnification of alizarin red staining of hDPSCs cultured in (control) 2D culture, on PLA, PLA/nHA/NAR, PLA/MTA, after incubation in cell culture media for 21 days. The scale bar in the control image is 30 µm and other images are 10 µm. (**B**) Bar chart showing photometric quantification of alizarin red for all groups. Different letters above bars represent statistical significance at *p* ≤ 0.05.

**Figure 5 polymers-17-00596-f005:**
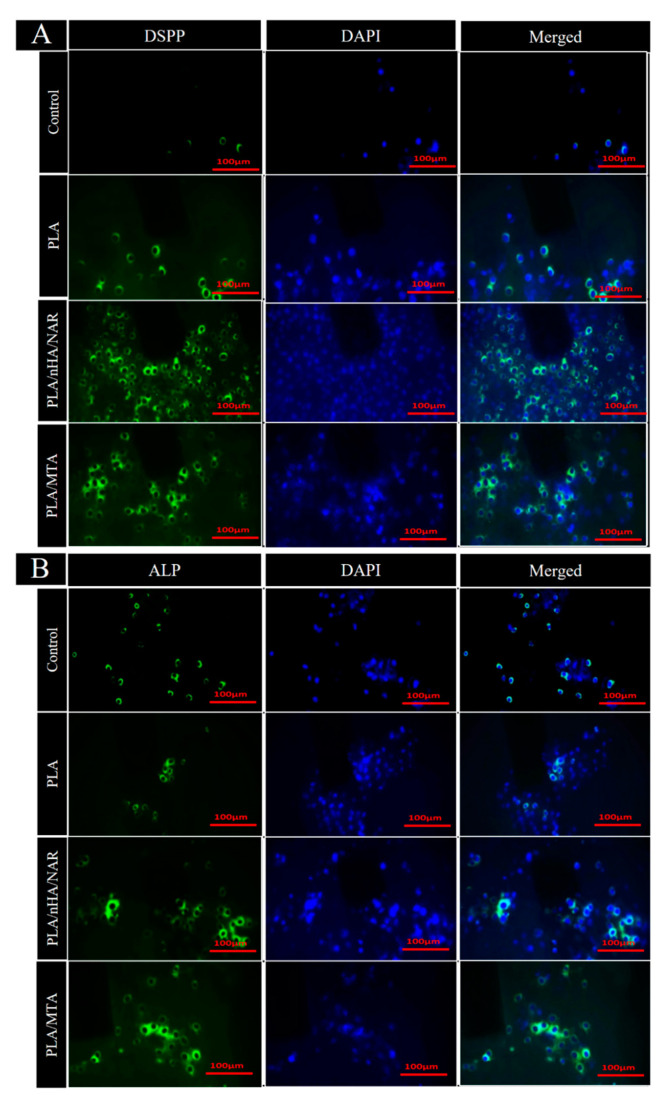
Immunofluorescence staining for DSPP (**A**) and ALP (**B**) for all groups (control, PLA, PLA/nHA/NAR, and PLA/MTA). Each group is represented by three images (antibody only, DAPI only, and composite image for both staining).

**Table 1 polymers-17-00596-t001:** Parameters used in the FDM printing process.

Parameter	Value
Bed temperature	40 °C
Nozzle temperature	210 °C
Printing speed	15 mm/s
Layer height	100 µm
Lay down pattern	0/90°

## Data Availability

The original contributions presented in this study are included in the article. Further inquiries can be directed to the corresponding author.

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
