# Peer review of "Inducing Osteogenesis in Human Pulp Stem Cells Cultured on Nano-Hydroxyapatite and Naringin-Coated 3D-Printed Poly Lactic Acid Scaffolds"

_polymers, 2025, doi:10.3390/polym17050596_

Round 1

Reviewer 1 Report

Comments and Suggestions for Authors

1.Introduction Section:

Flow and Structure: The introduction does a good job of setting the stage for the study, but it would benefit from more precise language in certain areas:

In the sentence "Tissue engineering is a dynamic vital process that aims to restore damaged, injured, or missing tissues," "dynamic vital process" is somewhat redundant. Consider revising to "Tissue engineering is a critical approach aimed at restoring damaged or missing tissues."

The discussion about the challenges of tissue engineering could be expanded to briefly mention some specific challenges faced in dental pulp regeneration, creating a clearer connection to the study’s objectives.

Literature Citations:

Some references (e.g., [1], [2], [3], etc.) are not complete and may benefit from additional context. For example, after citing "mesenchymal stem cells (MSCs) due to their self-renewability," it would be useful to mention how MSCs are pivotal in dental tissue engineering specifically.

Clarify the transition between discussing MSCs and the subsequent paragraph on PLA scaffolds. They are related, but the shift feels a bit abrupt.

2.Materials and Methods:

The cell culture protocol is described thoroughly, but it could be improved by clarifying the incubation conditions. For example, the phrase "the cells were expanded to passage 4" could be followed by a brief explanation of why passage 4 was chosen.

The methods for cell adhesion evaluation using SEM are good, but it might be worth noting whether cell viability was assessed as well.

3. Results and Discussion:

The results are well-organized, but some of the statistical analysis could be described more clearly. For instance, when comparing calcium deposition between the PLA/nHA/NAR and PLA/MTA scaffolds, it would be helpful to briefly explain the specific statistical test used (e.g., t-test, ANOVA) for the p-value reporting.

For the immunofluorescence analysis (ALP and DSPP), consider adding a figure that visually represents the fluorescence intensity of these markers across different groups. This would improve the clarity of the results and make the data more digestible.

The discussion could provide more insight into the comparative analysis of PLA/nHA/NAR scaffolds with other materials, particularly focusing on how this combination compares to existing treatment methods in regenerative dentistry. A deeper dive into why the combination of nHA and NAR leads to superior osteogenesis compared to MTA would be valuable.

There should be a mention of any potential limitations or challenges in the study (e.g., scalability of the 3D printing process, potential cytotoxicity of the coatings, etc.).

4. Current statement: "The attached cells appear spindle-shaped fibroblast-like appearance with long cytoplasmic processes."

Question: While this statement describes the cell morphology, it could be more clearly combined with the specific characterization of the cells in the SEM images and comparison with other groups.

Recommendation: “SEM images showed that the attached cells exhibited a spindle-shaped fibroblast-like appearance with long cytoplasmic processes, which was most prominent in the PLA/nHA/NAR group, suggesting that this combined surface was effective in promoting cell adhesion. It has better biocompatibility.”

5. Calcium concentration assay

Current statement: “The highest mean value of calcium deposition was observed for PLA/nHA/NAR, followed by PLA/MTA scaffold cultures...”

Issues: The biological significance of calcium deposition needs to be explained more clearly and the interpretation of statistical differences needs to be more refined.

Recommendation: “The amount of calcium deposition in the PLA/nHA/NAR group was significantly higher than that in the PLA/MTA group (p=0.012), indicating a stronger potential for bone mineralization and cell differentiation in this group. This result suggests that PLA / The bioactivity of the nHA/NAR combination in promoting osteo-tooth differentiation is more prominent. ”

6. Immunofluorescence staining

Current statement: "Immunoreactivity of these two markers is quite apparent in the cytoplasm of cells cultured PLA/nHA/NAR and PLA/MTA groups."

Question: The expression differences of immune markers (ALP and DSPP) between different groups can be further analyzed and combined with their significance for cell differentiation.

Recommendation: “The results of immunofluorescence staining showed that ALP and DSPP were significantly immunoreactive in the cytoplasm of the PLA/nHA/NAR and PLA/MTA groups, indicating that the expression of these two markers is related to the osteogenic/odontogenic processes of the cells. Compared with the control group and pure PLA group, the expression levels of ALP and DSPP in the PLA/nHA/NAR group were significantly higher (p<0.05), indicating that it has a role in promoting mineralization and dentin/bone differentiation. Stronger biological activity.”

7. Current statement: “Incorporation of NAR in the suggested formula has shown great potential in the proliferation and differentiation of hDPSCs.”

Question: The mechanism of action of NAR can be further elaborated based on the literature, especially its role in the Wnt/β-catenin pathway.

Recommendation: “Naringin (NAR), a natural flavonoid compound, has been shown to activate the Wnt/β-catenin signaling pathway, which plays an important regulatory role in bone and tooth differentiation. By activating this pathway, NAR It can upregulate the expression of osteogenic markers including DSPP, thereby enhancing the differentiation potential of hDPSCs. Literature shows that NAR regulates local pH through antioxidant effects, helps to increase the solubility of nHA, and then promotes the sustained release of calcium ions. This accelerates the bone formation process.”

8. Current statement: “Several limitations that might encountered within this in vitro study, as it may not fully replicate the complexity available within the in vivo models...”

Question: The limitations can be combined with future research directions to propose specific improvement suggestions.

Recommendation: "The limitation of this study is that it is limited to in vitro experiments and fails to fully simulate the complexity of the in vivo environment. Future studies can verify the long-term effects of the scaffold through animal models, especially in terms of blood flow, immune response, and tissue integration. Under the influence of biological factors, the feasibility and effectiveness of its clinical application are evaluated. "

Author Response

Response to Reviewer 1 Comments

The authors thank the reviewer for their productive and valuable comments about the manuscript. Your time and effort are sincerely appreciated. Below, you will find detailed, expanded responses to your comments, along with the corresponding revisions and corrections highlighted in the revised manuscript.

    Introduction Section:

Comments 1: Flow and Structure: The introduction does a good job of setting the stage for the study, but it would benefit from more precise language in certain areas.

Answer: The authors thank the reviewer for their comment regarding language of the introduction section. Accordingly, double-check has been done and some minor changes have been made and highlighted in the revised manuscript.

Comment 2: In the sentence "Tissue engineering is a dynamic vital process that aims to restore damaged, injured, or missing tissues," "dynamic vital process" is somewhat redundant. Consider revising to "Tissue engineering is a critical approach aimed at restoring damaged or missing tissues.

Answer: Authors thank the reviewer for their suggestion and the required change has been made

Comment 3: The discussion about the challenges of tissue engineering could be expanded to briefly mention some specific challenges faced in dental pulp regeneration, creating a clearer connection to the study’s objectives.

Answer: Authors thank the reviewer for their insightful comment and the changes have been made as required.

Literature Citation

Comment 4: Some references (e.g., [1], [2], [3], etc.) are not complete and may benefit from additional context. For example, after citing "mesenchymal stem cells (MSCs) due to their self-renewability," it would be useful to mention how MSCs are pivotal in dental tissue engineering specifically. Clarify the transition between discussing MSCs and the subsequent paragraph on PLA scaffolds. They are related, but the shift feels a bit abrupt.

Answer: Authors appreciate the time and effort of the reviewer to provide detailed comments. Accordingly, references have expanded and the transition between MSCs and PLA scaffolds has been smoothed up to improve clarity

  1. Material and Methods:

Comment 5: The cell culture protocol is described thoroughly, but it could be improved by clarifying the incubation conditions. For example, the phrase "the cells were expanded to passage 4" could be followed by a brief explanation of why passage 4 was chosen.

Answer: Authors appreciate the reviewer’s comment and the suggested amendment has been performed.

Comment 6: The methods for cell adhesion evaluation using SEM are good, but it might be worth noting whether cell viability was assessed as well."

Answer: Authors would like to thank the reviewer for his valuable feedback, Regarding the comment; while the cell viability assay associated with SEM was not conducted in this study, the data presented later in the manuscript strongly supports the viability and functionality of the cells attached to these scaffolds. The calcium concentration assay, positive staining for ALP (alkaline phosphatase) and DSPP (dentin sialophosphoprotein) osteogenic /odontogenic markers, confirm that the attached cells are not only viable but also metabolically active and expressing key markers associated with osteogenic and odontogenic differentiation.

  1. Results and Discussion:

Comment 7:The results are well-organized, but some of the statistical analysis could be described more clearly. For instance, when comparing calcium deposition between the PLA/nHA/NAR and PLA/MTA scaffolds, it would be helpful to briefly explain the specific statistical test used (e.g., t-test, ANOVA) for the p-value reporting.

Answer: The authors appreciate the reviewer’s comment regarding statistical analysis and a paragraph describing the statistical analysis was added to (page-7) at the end of materials and methods section accordingly.

Comment 8: For the immunofluorescence analysis (ALP and DSPP), consider adding a figure that visually represents the fluorescence intensity of these markers across different groups. This would improve the clarity of the results and make the data more digestible.

Answer: The authors thank the reviewer for their suggestion, however, the requested figure is already presented in Figure 5 (C and D) which illustrates the fluorescence intensities for ALP and DSPP respectively.  

Comment 9: The discussion could provide more insight into the comparative analysis of PLA/nHA/NAR scaffolds with other materials, particularly focusing on how this combination compares to existing treatment methods in regenerative dentistry.

Answer: Authors appreciate the reviewer’s comment. The suggested comparison has been addressed in the last paragraph of the discussion section before the study limitation paragraph.

Comment 10: A deeper dive into why the combination of nHA and NAR leads to superior osteogenesis compared to MTA would be valuable.

Answer: Authors appreciate the reviewer's suggestion. The suggested comparison is already discussed in the discussion section in a paragraph from lines 419 to 425.

Comment 11: There should be a mention of any potential limitations or challenges in the study (e.g., scalability of the 3D printing process, potential cytotoxicity of the coatings, etc.).

Answers: Authors appreciate the reviewer's comment. All suggestions have been addressed and highlighted in the limitation part of the revised manuscript.

Comment 12: 4. Current statement. "The attached cells appear spindle-shaped fibroblast-like appearance with long cytoplasmic processes" Question: While this statement describes the cell morphology, it could be more clearly combined with the specific characterization of the cells in the SEM images and comparison with other groups.

Recommendation: “SEM images showed that the attached cells exhibited a spindle-shaped fibroblast-like appearance with long cytoplasmic processes, which was most prominent in the PLA/nHA/NAR group, suggesting that this combined surface was effective in promoting cell adhesion. It has better biocompatibility.”

Answer: Authors would like to thank the reviewer for their insightful comment. The suggested recommendation has been taken into account and made the necessary adjustments, which have helped to improve the clarity and quality of the manuscript. (highlights in lines 287 to 291).

Comment 13: 5. Calcium concentration assay Comment: Current statement: “The highest mean value of calcium deposition was observed for PLA/nHA/NAR, followed by PLA/MTA scaffold cultures...”

Issues: The biological significance of calcium deposition needs to be explained more clearly and the interpretation of statistical differences needs to be more refined.

Recommendation: “The amount of calcium deposition in the PLA/nHA/NAR group was significantly higher than that in the PLA/MTA group (p=0.012), indicating a stronger potential for bone mineralization and cell differentiation in this group. This result suggests that PLA / The bioactivity of the nHA/NAR combination in promoting osteo-tooth differentiation is more prominent. ”

Answer: Authors thank the reviewer for their suggested amendment and the required change has been done and highlighted in the revised manuscript (highlights in lines 299 to 303).

Comment 14: 6. Immunofluorescence staining

Comment. Current statement: "Immunoreactivity of these two markers is quite apparent in the cytoplasm of cells cultured PLA/nHA/NAR and PLA/MTA groups."

Question: The expression differences of immune markers (ALP and DSPP) between different groups can be further analyzed and combined with their significance for cell differentiation.

Recommendation: “The results of immunofluorescence staining showed that ALP and DSPP were significantly immunoreactive in the cytoplasm of the PLA/nHA/NAR and PLA/MTA groups, indicating that the expression of these two markers is related to the osteogenic/odontogenic processes of the cells. Compared with the control group and pure PLA group, the expression levels of ALP and DSPP in the PLA/nHA/NAR group were significantly higher (p<0.05), indicating that it has a role in promoting mineralization and dentin/bone differentiation. Stronger biological  activity.”

Answer: Authors thank the reviewer for their suggested amendment and the required change has been done and highlighted in the revised manuscript (highlights in lines 324 to 331).

Comment 15: 7. Current statement: “Incorporation of NAR in the suggested formula has shown great potential in the proliferation and differentiation of hDPSCs.”

Question:The mechanism of action of NAR can be further elaborated based on the literature, especially its role in the Wnt/β-catenin pathway.

Recommendation: “Naringin (NAR), a natural flavonoid compound, has been shown to activate the Wnt/β-catenin signaling pathway, which plays an important regulatory role in bone and tooth differentiation. By activating this pathway, NAR It can upregulate the expression of osteogenic markers including DSPP, thereby enhancing the differentiation potential of hDPSCs. Literature shows that NAR regulates local pH through antioxidant effects, helps to increase the solubility of nHA, and then promotes the sustained release of calcium ions. This accelerates the bone formation process.”

Answer: Authors thank the reviewer for their suggested amendment and the required change has been done and highlighted in the revised manuscript (highlights in lines 398 to 407).

Comment 15: 8. Current statement “Several limitations that might encountered within this in vitro study, as it may not fully replicate the complexity available within the in vivo models...”

Question: The limitations can be combined with future research directions to propose specific improvement suggestions.

Recommendation: "The limitation of this study is that it is limited to in vitro experiments and fails to fully simulate the complexity of the in vivo environment. Future studies can verify the long-term effects of the scaffold through animal models, especially in terms of blood flow, immune response, and tissue integration. Under the influence of biological factors, the feasibility and effectiveness of its clinical application are evaluated. "

Answer: Authors thank the reviewer for their suggested amendment and the required change has been done and highlighted in the revised manuscript (Page13).

Reviewer 2 Report

Comments and Suggestions for Authors

This is an innovative study and I am happy to go through the revision later. But I have some major comments related to the methods section specifically:

1. Fabrication of the 3D Printed PLA Scaffold

I find this part quite confusing, the author should provide further details on:

The type of PLA filament (manufacturer, grade, molecular weight, and any preprocessing).

More  3D printing parameters including  infill density, extrusion rate

Why 300 µm pore size is used? Was this based on prior literature or pilot experiments?

The dip-coating process for the bioactive material can be expanded to include :

The thickness of the coating layers per dipping cycle;  coating uniformity test;

2. In Vitro Cell Culture

human dental pulp stem cells (hDPSCs) : passage number used for experiments should be given

The FBS pre-soaking step for scaffold conditioning :  its necessity and its influence on hydrophilicity should be discussed.

Trypsin-EDTA treatment for cell detachment: concentration and duration of trypsinization should be reported

3. Cell Adhesion and Morphology (SEM Analysis)

any additional post-coating characterization (e.g., FTIR, or contact angle measurements) was conducted to confirm successful bioactive incorporation.

Were any quantitative methods used for cell attachment assessment?

How were samples sputter-coated, and was a uniform gold layer applied to avoid imaging artifacts?

4. Osteogenic Differentiation and Mineralization Assays

(i) Calcium Concentration Assay

Pls provide: Calibration curve details (e.g., standard curve concentration range, R² value); the number of replicates used per sample to confirm statistical robustness; confirmation that no osteogenic differentiation medium was used (ensuring differentiation was scaffold-induced).

(ii) Alizarin Red Staining (ARS)

The extraction method using acetic acid should clarify incubation times to ensure standardization.

5. Immunofluorescence Staining for ALP and DSPP

The blocking step using 1% BSA in PBS :

-any negative control used to confirm specificity?

-The fluorescence intensity quantification using ImageJ should mention whether background subtraction was applied.

6. Statistical Analysis

The statistical test used should be stated in the methodology

Were data tested for normality (Shapiro-Wilk, Kolmogorov-Smirnov) before parametric tests were applied?

Were post-hoc tests applied to compare multiple groups?

Author Response

Response to Reviewer 2 Comments

The authors thank the reviewer for their productive and valuable comments about the manuscript. Your time and effort are sincerely appreciated. Below, you can find detailed, expanded responses to your comments, along with the corresponding revisions and corrections highlighted in the revised manuscript.

  1. Fabrication of the 3D Printed PLA Scaffold

 Comment: I find this part quite confusing, the author should provide further details on:

The type of PLA filament (manufacturer, grade, molecular weight, and any preprocessing).

More  3D printing parameters including  infill density, extrusion rate.

Why 300 µm pore size is used? Was this based on prior literature or pilot experiments?

The dip-coating process for the bioactive material can be expanded to include : The thickness of the coating layers per dipping cycle; coating uniformity test;

Answer: Authors thank the reviewers for their contribution to enhancing the quality of the manuscript.

Regarding your queries about the fabrication and modification of the 3D-printed PLA scaffold, I would like to clarify that the requested details, including the type of PLA filament (manufacturer, grade, molecular weight, 3D printing parameters (infill density and extrusion rate), the rationale for selecting a 300 µm pore size based on previous publication, as well as specifics, characterizations about the dip-coating process are comprehensively addressed in our previous publication:

 "Fabrication and characterization of 3D-printed polymeric-based scaffold coated with bioceramic and naringin for a potential use in dental pulp regeneration (DOI: 10.1111/iej.14194).

This previous paper has been cited as a reference in the current manuscript (Reference no 17). This work serves as the foundation for the methodologies and characterizations employed in the current manuscript.

2: In Vitro Cell culture

Comment:  human dental pulp stem cells (hDPSCs) : passage number used for experiments should be given.

Answer: Authors thank the reviewer for their comment. Authors would like to kindly point out that the specific passage number is already addressed in the manuscript on page 4 (line 139). However, I will ensure that this information is clearly highlighted to avoid any potential ambiguity.

Comment: The FBS pre-soaking step for scaffold conditioning:  its necessity and its influence on hydrophilicity should be discussed.

Answer:

Thank you for your insightful comments on our manuscript. We appreciate your suggestion regarding discussing the importance and influence of FBS pre-soaking on hydrophilicity of the scaffold. According to Vetsch et al. (2015) in their article "Effect of fetal bovine serum on mineralization in silk fibroin scaffolds," published in Acta Biomaterialia, fetal bovine serum contains essential components such as growth factors and proteins that are crucial for cell proliferation and attachment. Pretreating the scaffold with FBS enhances cell attachment due to the presence of these nutritional constituents and macromolecular entities like amino acids, carbohydrates, lipids, and hormones, all of which contribute to its biological functionality. This reference has been cited in the manuscript to support the choice of using FBS for cell attachment. To ensure clarity and provide readers with a comprehensive understanding

Comment: Trypsin-EDTA treatment for cell detachment: concentration and duration of trypsinization should be reported.

Answer: Authors thank the reviewer’s feedback and suggestions. Regarding  the comment about Trypsin-EDTA treatment for cell detachment. The required amendment has been made and highlighted in (page 4)  of the revised manuscript.

3) Cell Adhesion and Morphology (SEM Analysis)

Comment: Any additional post-coating characterization (e.g., FTIR, or contact angle measurements) was conducted to confirm successful bioactive incorporation.

Answer: Authors would like to thank the reviewer for their valuable insight regarding post coating characterization and would like to clarify that in our previous study (Fabrication and characterization of 3D- printed polymeric- based scaffold coated with bioceramic and  naringin for a potential use in dental pulp regeneration (in vitro study). Fourier-transform infrared spectroscopy where (FTIR) was conducted as part of the post-coating characterization to confirm successful bioactive incorporation, and functional group incorporation.  Additionally, comprehensive characterization was performed using scanning electron microscopy (SEM), and energy-dispersive X-ray spectroscopy (EDX) to analyze surface morphology, and elemental composition.(Reference no.17)

Comment. Were any quantitative methods used for cell attachment assessment?

Answer: The authors thank and appreciate the reviewer's query regarding quantitative methods for assessing cell attachment. In this study, we did not employ quantitative cell attachment assays, as our primary goal was to evaluate the level of cell adhesion qualitatively through scanning electron microscopy (SEM) analysis. As this approach was chosen to evaluate cell morphology and adherence patterns directly, which aligned with the objectives of our  research.

Comment. How were samples sputter-coated, and was a uniform gold layer applied to avoid imaging artifacts.

Answer: Authors appreciate the reviewer for their insightful query. The required amendment has been made and highlighted in the revised manuscript in (page 4-5) (164-168).

  1. Osteogenic Differentiation and Mineralization Assays

(1) Calcium Concentration Assay

 Comment:  Pls provide: Calibration curve details (e.g., standard curve concentration range, R² value)

Answer:  Authors would like to thank the reviewers for their comment regarding calibration curve.

We would like to clarify that it was added to the revised manuscript (Figure.5B) at (page 11).

Comment:  the number of replicates used per sample to confirm statistical robustness.

Answer: Authors would like to thank the reviewers for their comment regarding the number of replicates. Authors would like to clarify that, the assay was conducted in triplicates (n=3), to ensure statistical robustness throughout multiple measurements across the groups. This point has been added, and highlighted on page 5 in the revised manuscript.

Comment:  confirmation that no osteogenic differentiation medium was used (ensuring differentiation was scaffold-induced)

Answer: Authors thank the reviewer for their feedback. A confirmation note was added, and highlighted in the revised manuscript, stating no osteogenic medium was used during the experiment, at page 5 of the manuscript (175-177)

 (2) Alizarin Red Staining (ARS)

Comment: The extraction method using acetic acid should clarify incubation times to ensure standardization.

Answer: Authors would like to thank the reviewer for their comment regarding acetic acid incubation times. The necessary changes have been made and highlighted in page 6 (232-234).

  1. Immunofluorescence Staining for ALP and DSPP

Comment: The blocking step using 1% BSA in PBS :

Answer: Authors thank the reviewer their comment and the required amendment has been highlighted in the revised manuscript.

Comment: any negative control used to confirm specificity?

Answer: Authors thank and appreciate the reviewers suggestion regarding using negative control groups, we would like to clarify that the nature of our experiment focuses on comparing different scaffold treatment conditions against untreated cell cultures and neat PLA scaffold without any treatment. Comparisons among these groups may serve as baseline reference, that effectively highlight differences in protein expression among different treatment conditions, besides staying relevant with our experimental framework   

Comment The fluorescence intensity quantification using ImageJ should mention whether background subtraction was applied.

Answer: The authors would like to thank the reviewer for raising this important point regarding background fluorescence. The required change has been done and highlighted in the corresponding section in (page 6 )of the revised manuscript (261-264)

  1. Statistical Analysis

The statistical test used should be stated in the methodology

Were data tested for normality (Shapiro-Wilk, Kolmogorov-Smirnov) before parametric tests were applied?

Were post-hoc tests applied to compare multiple groups?

Answer: Authors would like to thank the reviewer for their feedback related to statistical analysis stated in the methodology. A detailed statistical section has been added at the end of methodology section highlighting all required information (page 7) (269-279)

Round 2

Reviewer 1 Report

Comments and Suggestions for Authors

Accept in present form